# Current Therapeutic Strategies in Patients with Oropharyngeal Squamous Cell Carcinoma: Impact of the Tumor HPV Status

**DOI:** 10.3390/cancers13215456

**Published:** 2021-10-29

**Authors:** Alexandre Bozec, Dorian Culié, Gilles Poissonnet, François Demard, Olivier Dassonville

**Affiliations:** Institut Universitaire de la Face et du Cou, Centre Antoine Lacassagne, Université Côte d’Azur, 06100 Nice, France; dorian.culie@nice.unicancer.fr (D.C.); gilles.poissonnet@nice.unicancer.fr (G.P.); francois.demard@nice.unicancer.fr (F.D.); olivier.dassonville@nice.unicancer.fr (O.D.)

**Keywords:** oropharynx, neoplasm, squamous cell carcinoma, human papillomavirus, therapeutic management, treatment selection

## Abstract

**Simple Summary:**

Contrary to other head and neck subsites, oropharyngeal squamous cell carcinoma (OPSCC) has shown a considerable increase in incidence over the past 20 years. This growing incidence is largely due to the increasing place of human papillomavirus (HPV)-related tumors. HPV-positive and HPV-negative OPSCC are two distinct entities with considerable differences in terms of treatment response and prognosis. However, there are no specific recommendations yet in the therapeutic management of OPSCC patients according to their tumor HPV-status. The aim of this review is therefore to discuss the therapeutic management of patients with OPSCC and the impact of HPV status on treatment selection.

**Abstract:**

Since there is no published randomized study comparing surgical and non-surgical therapeutic strategies in patients with oropharyngeal squamous cell carcinoma (OPSCC), the therapeutic management of these patients remains highly controversial. While human papillomavirus (HPV)-positive and HPV-negative OPSCC are now recognized as two distinct diseases with different epidemiological, biological, and clinical characteristics, the impact of HPV status on the management of OPSCC patients is still unclear. In this review, we analyze the current therapeutic options in patients with OPSCC, highlighting the most recent advances in surgical and non-surgical therapies, and we discuss the impact of HPV status on the therapeutic strategy.

## 1. Introduction

Head and neck squamous cell carcinoma (HNSCC) accounts for more than 600,000 new cases each year worldwide and represents the 6th cause of cancer deaths [1,2,3]. In western countries, about 25% of all HNSCC arise from the oropharynx [2]. Beside alcohol and tobacco consumption, which are well-known risk factors, human papillomavirus (HPV) has been also implicated in the carcinogenesis of oropharyngeal squamous cell carcinoma (OPSCC) [1]. Contrary to other head and neck subsites, OPSCC incidence has considerably increased in the past 20 years [2,4]. This growing incidence is largely due to the increasing place of HPV-related OPSCC [2,4]. Indeed, to date, HPV-positive OPSCC represents up to 80% of OPSCC in North America and Northern Europe, and 30 to 60% in Western Europe [5,6]. HPV-positive and HPV-negative OPSCC are two distinct entities with molecular, histological, epidemiological, and prognostic differences [7]. However, there are no specific recommendations yet in the therapeutic management of patients with OPSCC according to their tumor HPV-status [8,9].

There has been considerable debate over the last three decades regarding the initial therapeutic management of OPSCC [10,11]. To date, there is no published prospective randomized clinical trial comparing surgical and non-surgical approaches for OPSCC patients. It is therefore widely acknowledged that the therapeutic decision has to be made by a tumor board, where the cases of individual cancer patients are thoroughly reviewed by a team of physicians and other health professionals from different specialties (surgeons, medical and radiation oncologists, radiologists, and pathologists). This results in great variability in the therapeutic management of patients with OPSCC between medical teams, according to their particular experience and skills [11].

In the light of the results of larynx preservation programs, the 1990s saw an increased use of non-surgical treatments, for OPSCC, combining radiotherapy (RT) with chemotherapy (CT), commonly referred to as an organ-sparing therapy protocol [10,12]. This switch from primary conventional open surgery to definitive (chemo)-RT ((C)RT) was made in many centers despite the lack of high-quality evidence from randomized controlled trials in patients with OPSCC. In the meantime, HPV-positive OPSCC has been identified as a unique disease, with improved radiosensitivity and survival [13]. Thus, the favorable outcomes reported with non-surgical therapeutic strategies in previous North American and European studies could be explained by a high yet unknown proportion of HPV-related OPSCC [14]. Recently, the presumed equivalence in terms of oncological results between surgical and non-surgical therapeutic strategies in patients with HPV-negative OPSCC was rediscussed, because the HPV status of OPSCC patients was not determined before 2010 and could be a major bias in earlier studies [15]. Simultaneously, the development of minimally-invasive surgical procedures and the progress in microvascular reconstructive surgery have considerably reduced the classic sequelae of oropharyngeal oncologic surgery [16,17]. Altogether, these data explain why the role of upfront surgery and the impact of tumor HPV status in the initial management of OPSCC remain largely debated.

The aim of this review article is therefore to discuss current therapeutic strategies in patients with OPSCC and the potential impact of tumor HPV status.

## 2. HPV-Positive and HPV-Negative OPSCC Are Two Distinct Diseases

HPV-positive OPSCC is a unique entity both clinically and demographically [2,7]. HPV-positive OPSCC patients displayed less comorbidity, less alcohol and tobacco consumption, higher educational level and socio-economic status, lower T stage, higher N stage, and more frequent involvement of the tongue base and tonsillar fossa than HPV-negative OPSCC patients [11,18]. Moreover, recent studies showed that the risk of synchronous or metachronous second primary neoplasia was significantly reduced in HPV-negative OPSCC patients [19,20,21].

At the molecular level, HPV-induced carcinogenesis leads to functional abrogation of p53 and pRb pathways, mediated by the expression of the viral oncoproteins E6 and E7 [22]. E6 binds wild-type p53 and induces its degradation, leading to impaired apoptosis. E7 binds pRb, causing the release of the transcriptional factor E2F that activates cellular proliferation [22]. Independently of pRb inhibition, the transcriptional induction of KDM6B by E7 accounts for expression of the p16 protein, an inhibitor of cyclin-dependent kinases (CDK) 4 and 6 [23]. Consequently, contrary to their HPV-negative counterparts, HPV-positive OPSCC bear high p16 levels, and the overexpression of p16 is used in routine clinical practice as a surrogate marker of tumor HPV status [24]. However, although p16 overexpression is a cost-effective and practical marker of HPV-positive OPSCC, the link between p16 overexpression and HPV-induced carcinogenesis is not totally specific, and 5 to 20% of p16-positive OPSCC are not HPV-related [24,25].

In a comprehensive genomic characterization of HNSCC, the Cancer Genome Atlas Network showed that HPV-associated tumors are dominated by helical domain mutations of the oncogene PIK3CA, novel alterations involving loss of TRAF3, and amplification of the cell cycle gene E2F1 [26]. By contrast, smoking-related HNSCCs demonstrate near universal loss-of-function TP53 mutations and CDKN2A inactivation with frequent copy number alterations including amplification of 3q26/28 and 11q13/22 [26].

Regarding clinical presentation, HPV-positive OPSCC are characterized by a frequent discordance between small primary tumor size and significant lymph node involvement [27]. This explains that an isolated neck mass (carcinoma of unknown primary: “CUP” syndrome) is a common initial presentation of HPV-positive OPSCC. Neck metastases are often cystic and, therefore, solitary cystic metastatic lymph node of occult HPV-positive OPSCC can mimic a second branchial cleft cyst [28]. Moreover, the primary tumor can be difficult to see and to delineate in the lymphoepithelial tissue of the tongue base or tonsillar fossa.

Multiple studies demonstrated that HPV tumor status was the main prognostic factor in OPSCC patients [6,7,29]. The improved prognosis for HPV-positive OPSCC can be explained by several factors. Firstly, due to lower alcohol/tobacco consumption, patients with an HPV-positive OPSCC display a lower level of comorbidity than patients with an HPV-negative OPSCC. This better general health status makes HPV-positive OPSCC patients more likely to benefit from the treatment at any stage of the disease [27].

Secondly, HPV-positive OPSCC are characterized by an improved chemo- and radiosensitivity compared to HPV-negative OPSCC [29,30]. Complete response rates after CRT are considerably higher for HPV-positive than for HPV-negative patients [29]. Moreover, the risk of locoregional or distant recurrence is significantly lower for HPV-positive than for HPV-negative patients irrespective of the therapeutic strategy (surgical or non-surgical treatment) [31,32]. The pattern of recurrence is also different according to the HPV tumor status. Indeed, a recent multicentric study showed that locoregional recurrence was the most frequent type of treatment failure in HPV-negative patients, whereas distant metastasis was the main type of recurrence in HPV-positive patients [31]. In HPV-positive OPSCC, the risk of locoregional recurrence is inferior to 15% and most locoregional recurrences correspond to nodal persistent/recurrent disease frequently amenable to salvage neck dissection [31]. At the opposite, locoregional failures in HPV-negative OPSCC patients are most often local recurrence or combine local with nodal recurrences and are rarely amenable to successful surgical salvage [31,33].

Thirdly, HPV-positive OPSCC patients display a lower risk of second primary neoplasia than HPV negative OPSCC patients [19,20]. The significant risk of second cancer in HPV-negative patients is mainly explained by their alcohol/tobacco consumption with the concept of field cancerization and affects mostly the head and neck, the lung, and the esophagus [19,34]. It is considered to be one of the leading causes of death in patients that have been cured from their primary OPSCC [19]. The HPV oncogenic properties at other cancer sites and in particular the anogenital organs are well demonstrated [35,36]. In this regard, in a recent study investigating sequential acquisition of HPV infection between genital and oral anatomic sites in males, Dickey et al. showed that the Hazard ratio of a sequential genital to oral HPV infection was 2.3 (95% CI: 1.7–3.1) and 3.5 (95% CI: 1.9–6.4) for oral to genital infection [35]. However, the risk of a second HPV-induced primary malignancy seems relatively low and does not represent an important cause of death in HPV-positive OPSCC patients [19,21].

All these differences and particularly the considerable discrepancy in terms of prognosis have led to the creation of two distinct TNM classifications, in the 8th UICC/AJCC staging system, according to the p16 tumor status of OPSCC patients [13,37]. The most important change for p16-positive OPSCC concerns nodal staging where clinically involved lymph nodes, whether one or multiple, as long as they are ipsilateral and less than 6 cm in size, are included in the same N category—N1—since there is no significant impact on survival. Survival with clinically palpable and/or radiographically evident, bilateral, or contralateral lymph nodes was distinguishable, with a worse outcome than N1. Therefore, contralateral or bilateral lymph nodes are classified as N2, without, conversely to p16-negative OPSCC, the classical three N2 sub-stages (N2a, b, or c) [13]. Of note, extranodal extension is not considered as a staging criterion for p16-positive OPSCC. Finally, the 8th edition staging of p16-positive OPSCC gives a much more accurate and reasonable prediction of survival [13]. For example, a patient that presents with a 2 cm p16-positive tonsil cancer and 2 metastatic neck lymph nodes in the same side was stage IV in the 7th edition staging manual but is stage I in the 8th edition. The psychological benefit of having stage I vs. stage IV cancer is significant and means that clinicians can much more readily reassure patients that they have a good prognosis. However, currently, this new classification should not be used to modify the therapeutic strategy, and, in particular, the de-escalation of treatment intensity for p16-positive patients should only be tested in clinical trials [38].

Interestingly, several studies demonstrated that, besides p16 tumor status, tobacco consumption was an important prognostic factor in OPSCC patients [39,40]. Indeed, p16-positive OPSCC occurring in smokers (>10 pack-years) exhibit an intermediate prognosis between p16-positive tumors in non-smokers, which are associated with the best prognosis, and p16-negative tumors, which are associated with the worst prognosis [41]. Therefore, p16-positive OPSCC occurring in smokers represents a complex phenomenon where the role of HPV and tobacco in the carcinogenic process is difficult to evaluate, and where the tumor biology can mix together genetic alterations induced by HPV and by tobacco consumption [42]. This type of tumor does not represent a rare situation, particularly in Latin European countries, where tobacco consumption is still frequent and where HPV-positive OPSCC have experienced a drastic rise over the past 20 years [27]. At the time of personalized medicine, advances in molecular characterization of the tumor could make it possible, in the near future, to precisely assess the specific prognosis of each OPSCC patient. Table 1 summarizes the main clinical characteristics of HPV-positive and HPV-negative OPSCC.

## 3. Standard Therapeutic Options in OPSCC Patients

American and European guidelines on the management of OPSCC patients do not differ according to the HPV status of the tumor [8,43,44]. Indeed, two therapeutic options can be considered: one based on upfront surgery with or without adjuvant (C)RT, the other on definitive (C)RT. Early-stage (T1–T2, N0) OPSCCs can be managed by either primary surgery or definitive RT alone. Locally advanced (T3–T4, N0 and T1–T4, N1–N3) OPSCCs require multimodal therapy, including upfront surgery followed by RT or CRT according to pathological findings (surgical margins, extranodal extension), or definitive CRT.

### 3.1. Surgical Treatment

#### 3.1.1. Evolution of Surgical Techniques

OPSCC oncologic surgery has undergone an intense transformation over the past 30 years. Although complete tumor removal with free surgical margins remains the cornerstone of surgical treatment, surgical techniques have evolved from radical non-conservative open surgery to functional organ preservation surgery [45,46,47]. Since the beginning of the 2000s, the classical transmandibular approach has been replaced, in most cases, by a double transoral and transcervical approach, preventing the complications related to mandibular osteotomy [45]. Considerable efforts have been made to avoid the functional sequelae of pharyngotomy by expanding the indications for exclusive transoral approach in oropharyngeal oncologic surgery [12,48]. These advances have been made possible by the development of appropriate retractors and specific instruments as well as by the use of CO_2_ laser and especially robotic surgery (da Vinci^®^ Surgical System) [49,50]. Transoral robotic surgery (TORS) for OPSCC has been used mainly for small T1–T2 primary tumor of the tonsils or the tongue base and has demonstrated satisfactory oncologic outcomes with a rate of free surgical margins comparable to conventional open surgery [49]. Some authors have shown that TORS could also achieve complete resection of locally advanced OPSCC in selected cases with the most experienced TORS surgeons [48].

Similar to primary tumor resection, neck surgical treatment has evolved into more conservative procedures, from radical neck dissection to modified and selective neck dissection [51]. With the development of minimally-invasive surgical techniques and similarly to oral cavity cancers, the sentinel node biopsy (SNB) has been developed in patients with early-stage OPSCC and N0 neck [47]. In this regard, a recent randomized trial comparing SNB and neck dissection in T1–T2, N0 oral, and oropharyngeal cancer demonstrated the oncologic equivalence of the two approaches, with lower morbidity in the SNB arm during the first 6 months after surgery [52].

Along with the development of more conservative surgical procedures, considerable advances have been made in head and neck reconstructive surgery. This progress is mainly linked to the development and refinements of microvascular surgical techniques [16,53,54]. The fibular free-flap is considered as the gold standard for mandibular reconstruction, and the development of three-dimensional preoperative virtual planning has considerably facilitated the flap-shaping process, which was recognized as a critical point of this particularly complex surgical procedure [55,56]. Radial forearm and anterolateral thigh free-flaps are the two main technical options for reconstruction of large oropharyngeal soft-tissues defects [16]. Anterolateral thigh flap thinning techniques have enabled its use, as the radial forearm flap, when a thin and pliable flap is needed, with less morbidity to the donor site [53].

#### 3.1.2. Therapeutic Strategies Involving Upfront Surgery

Early-stage OPSCC (T1–T2, N0) can be managed by either surgery or RT alone [8,44]. When primary surgery has been performed, indication for adjuvant therapy is based on pathological findings. Surgery alone is sufficient after complete surgical resection of pT1–T2, N0 OPSCC, with free surgical margins (>5 mm) and without deleterious pathological features such as perineural invasion and vascular embolism. Postoperative RT has to be considered in cases of close surgical margins (1 to 5 mm), lymph node metastases, or perineural/vascular invasion. Postoperative concurrent CRT is indicated in patients less than 70 years of age with positive surgical margins or extranodal extension [57].

In patients with locally advanced (T3–T4, N0 and T1–T4, N1–N3) OPSCC treated by upfront surgery, postoperative RT is almost always indicated. A low proportion of patients with a pT3N0 tumor or with a small unique metastatic lymph node in the upper neck without any other adverse pathological feature may avoid postoperative RT [8,44]. Postoperative concurrent CRT is indicated in patients less than 70 years of age with positive surgical margins or extranodal extension [57]. Postoperative concurrent CRT should also be discussed in patients less than 70 years of age with close surgical margins, multiple lymph node metastases, and/or perineural/vascular invasion [57]. Although there is a large international consensus in favor of adapting cervical lymph node treatment to the primary therapy modality chosen for the primary tumor, primary upfront neck dissection before RT or CRT may be of interest in selected patients with small primary tumors but bulky nodal disease [10,58]. Indeed, although they have found no clear benefit in terms of overall survival (OS), some studies have shown improved disease-specific survival (DSS) and regional control rates with such a therapeutic strategy [59,60].

### 3.2. Non-Surgical Therapeutic Strategies

Non-surgical treatment of patients with OPSCC is mainly based on definitive RT. RT alone on the primary tumor and neck is sufficient for early-stage OPSCC, whereas it should be potentiated by a systemic therapy for locally advanced OPSCC [8,44].

Similar to head and neck surgery, RT techniques have largely evolved over the past 20 years. The use of intensity-modulated RT (IMRT) has become the gold standard in patients with OPSCC and has demonstrated an improved preservation of salivary function compared with conventional RT, without compromising locoregional control or survival [61]. IMRT also appears to have a favorable impact on swallowing and quality of life (QoL) outcomes [62]. With the rapid advances in medical physics, RT techniques are constantly evolving. Further technical refinements in RT of head and neck tumor and particularly of OPSCC have been recently evaluated and are currently being tested in clinical trials (Helical Tomotherapy, Volumetric Vodulated Arc Therapy, stereotactic RT, and proton RT) [63].

Three-weekly (100 mg/m²) cisplatin-based concurrent CRT is the gold standard non-surgical treatment of locally advanced OPSCC [8,63]. In patients who are not fit enough to receive this standard therapeutic regimen, weekly cisplatin-based CRT has been shown to be a reasonable alternative [64]. Since the results of Bonner’s study, cetuximab-based RT has been recognized as a valid therapeutic option in patients with locally advanced HNSCC [65]. A direct comparison with the standard of care, i.e., cisplatin-based CRT, was not available since the results of de-escalation prospective trials on HPV-positive OPSCC. These studies have shown that, when tested head-to-head, cisplatin was far more effective in terms of OS and locoregional control, with a different profile of toxicity but surprisingly comparable rates of grade 3 or 4 toxicity (grade 3: severe but not immediately life-threatening adverse event; grade 4: life-threatening adverse event according to the Common Terminology Criteria for Adverse Events-CTCAE v4.0) [38,66]. Therefore, cetuximab-based RT should be reserved to patients that are unfit for cisplatin-based CRT.

In contrast to de-escalation studies conducted in HPV-positive OPSCC, other studies have examined the opportunity to enhance treatment intensity by combining additional therapy with the conventional cisplatin-based CRT [67,68]. These studies did not produce encouraging results, which explains that cisplatin-based concurrent CRT remains the standard of care even for patients with unresectable OPSCC [8]. Indeed, the RTOG 0522 randomized phase III trial compared concurrent accelerated cisplatin-based CRT with or without cetuximab in patients with locally advanced HNSCC (70% of OPSCC, 73% of which were p16-positive) [68]. The study showed that adding cetuximab to cisplatin-based RT did not improve oncologic outcomes (3-year progression-free survival: PFS and OS, locoregional failure, and distant metastasis) but resulted in more grade 3 to 4 acute toxicities [68]. Two phase III randomized trials compared definitive CRT to induction CT (TPF: docetaxel, cisplatin, 5-fluorouracil, or PF) followed by CRT but failed to show any advantage of induction CT plus CRT over CRT alone [67,69]. Other prospective randomized studies have compared induction CT (TPF) followed by cetuximab-based RT to definitive cisplatin-based CRT but did not show any survival differences between the two therapeutic approaches [70]. Several studies are currently investigating the role of immunotherapy (anti-PD1/PD-L1 antibodies) added before or concomitantly to conventional CRT [71]. Although final data have yet to be released, preliminary results of the phase III JAVELIN head and neck 100 study, which were presented in the 2020 ESMO virtual congress, did not demonstrate statistically significant improvement in PFS with avelumab plus CRT compared with placebo plus CRT [71].

## 4. Impact of the HPV Tumor Status on Therapeutic Strategy

### 4.1. Early-Stage OPSCC

Early-stage (T1–T2, N0) OPSCC can be treated by either surgery or definitive RT alone. There is no demonstrated survival advantage with primary surgery compared to definitive RT for early-stage HPV-positive OPSCC [72,73]. Indeed, in a recent retrospective multicentric study conducted by the GETTEC collaborative study group, Culie et al. showed no significant differences in oncologic outcomes (OS, DSS, and recurrence-free survival: RFS) between the surgical and non-surgical treatment groups in 44 early-stage p16-positive OPSCC patients [72]. In a prospective phase II study (ORATOR) on 68 patients randomly assigned to primary TORS or definitive RT, Nichols et al. demonstrated no significant difference in OS or RFS between the two therapeutic strategies, but the trial was not designed for this purpose [73]. Despite a lower rate of tinnitus, hearing loss and neutropenia in patients receiving upfront surgery, one-year swallowing-related QoL scores were higher in patients treated by primary RT, although this improvement did not represent a clinically meaningful change [73].

Whatever the therapeutic strategy, the overall prognosis of these patients is excellent and the main objective should be therefore to preserve functions and QoL. Consequently, primary surgery should not be the first therapeutic option if the surgeon is not confident that it will provide optimal functional results. This depends mainly on the anatomical subsite and extension of the tumor. If surgery is selected, a transoral approach should be preferred. Indeed, minimally invasive surgery (TORS and elective neck dissection or SNB) has demonstrated promising results in terms of swallowing function and could be an interesting option to avoid late side-effects of RT [74,75]. Several phase III clinical trials (Best Of and TORPHYNX; ClinicalTrials.gov Identifiers: NCT02984410 and NCT04224389, respectively) comparing TORS and RT in terms of swallowing function in patients with early-stage OPSCC are currently ongoing. Hence, for patients with early-stage HPV-positive OPSCC, given that similar oncologic and functional results are obtained regardless of the therapeutic strategy, discussions between patients and tumor board (surgeon, radiation oncologist) remain the gold standard of the therapeutic decision-making process.

At the opposite, several retrospective studies showed that, for HPV-negative OPSCC, upfront surgery was associated with improved oncologic outcomes, including for early-stage tumors [6,15]. Indeed, in a retrospective multicentric analysis of 103 p16-negative early-stage OPSCC patients, Culie et al. showed significantly higher OS and DSS rates in patients treated by upfront surgery compared to those treated by definitive RT [15]. Similarly, the multicentric Papillophar French study showed that upfront surgery was independently associated with an improved PFS compared to non-surgical treatment in HPV-negative OPSCC patients [6]. Indeed, even after multivariate analysis taking into account performance status, alcohol/tobacco consumption, tumor stage, HPV status, and type of treatment, definitive RT was associated with worse OS (HR = 1.88; 95% CI: 1.10–3.21) and PFS (HR = 1.86; 95% CI: 1.19–2.92) [6]. Moreover, the possibility offered by primary surgery to reserve RT in case of second primary tumor has to be considered, particularly in patients with high alcohol/tobacco consumption [19]. Altogether, these data support the use of primary surgery in patients with early-stage HPV-negative OPSCC [10]. Figure 1 summarizes the therapeutic strategy for patients with early-stage OPSCC.

### 4.2. Locally Advanced Resectable OPSCC

In locally advanced (T3–T4, N0 and T1–T4, N1–N3) resectable OPSCC, primary surgery is associated with significant functional impairments depending on the anatomical subsite and tumor extensions [16,76]. Although TORS can be used in carefully selected cases (T1–T2 primary tumors, selected T3 primary tumors with adequate tumor exposition, and experienced robotic surgeon), most patients will require open surgery using a combined transoral/transcervical, or more rarely a transmandibular approach [10,45]. In most cases, a microvascular free-flap reconstruction will also be necessary. Since postoperative adjuvant therapy is required, upfront surgery will not prevent the use of RT. Compared with definitive CRT, primary surgery followed by RT or CRT is associated with longer treatment duration, higher costs, and possibly greater functional impairment [77]. Therefore, primary surgery should only be used if this therapeutic strategy is likely to improve survival outcomes.

In HPV-positive locally advanced OPSCC, there is no clear benefit in terms of survival of primary surgery followed by adjuvant (C)RT compared with definitive CRT [6,72,78]. Indeed, the French Papillophar study showed similar OS and PFS rates for HPV-positive OPSCC, whatever the primary treatment modality [6]. Similarly, Kelly et al., in a study investigating the outcomes of each treatment strategy using the American National Cancer Database, found that upfront surgery followed by adjuvant (C)RT and definitive CRT yielded comparable 3-year OS rates [79]. In the retrospective multicentric study of Culie et al. involving 338 p16-positive locally advanced OPSCC patients, there was no significant difference in OS between the surgical and non-surgical therapeutic strategies (5-year OS rates of 93.7% and 87.8%, respectively, *p* = 0.10) [72]. Of note, RFS was significantly higher in the surgical group than in the non-surgical group of patients (5-year RFS rates of 81.3% and 69.6%, respectively, *p* = 0.002). However, the high rate of successful surgical salvage for locoregional recurrence in the non-surgical group (17 successful surgical salvages for 26, local/regional recurrences) explained the absence of significant impact on OS [72]. Considering the lack of survival advantage, the additional costs, and the potential additional functional impairments, most patients with HPV-positive locally advanced OPSCC should be referred to definitive CRT, with surgery being reserved as a salvage procedure [10].

Conversely, although there is no randomized controlled study comparing surgery followed by adjuvant (C)RT and definitive CRT in patients with HPV-negative locally advanced OPSCC, most recent cohort studies suggested that primary surgery provided a clear benefit in terms of oncologic outcomes in this population [6,15,80]. Despite significant functional impairment, long-term clinical outcomes and QoL are acceptable given the survival advantage [16,81]. Indeed, in a recent study evaluating OS from the American National Cancer Database and involving 6,872 locally advanced OPSCC patients with a documented HPV status, Kamran et al. showed that patients treated with primary surgery followed by adjuvant (C)RT have improved survival compared with those treated with definitive CRT (HR = 0.79, 95% CI: 0.69–0.91, *p* = 0.001 for the whole cohort; HR = 0.74, 95% CI: 0.60–0.91, *p* = 0.005 in the HPV-negative group; and HR = 0.85, 95% CI: 0.70–1.03, *p* = 0.098 for the HPV-positive group) [80]. Similarly, in a prospective follow-up of 340 OPSCC patients with a previously determined HPV status, the French Papillophar study reported by Lacau St Guily et al. found, after multivariate analysis, a benefit in terms of OS and PFS for upfront surgery [6]. In the HPV negative cohort, 2-year PFS rates were 64% and 42% in the surgical and non-surgical cohorts, respectively [6]. In a retrospective multicentric analysis of 371 patients with a p16-negative locally advanced OPSCC, Culie et al. showed that upfront surgery was significantly associated with improved OS (*p* = 0.01), DSS (*p* = 0.02), and RFS (*p* = 0.02), compared with non-surgical treatment (5-year OS: 71.9 vs. 46.5%; DSS: 76.8 vs. 57.7%; RFS: 60.2 vs. 42.2%) [15]. In another retrospective study involving 3674 patients with an HPV-negative stage III–IVa (T1–2, N1–2b, and M0) OPSCC from the American National Cancer Database and Surveillance, Epidemiology, and End Results (SEER) program between 2010 and 2016, Jacobs et al. showed that, on weighted multivariable Cox regression, patients recommended to receive frontline surgery had improved OS compared with those recommended to receive CRT alone (HR = 0.77; 95% CI: 0.68–0.86) [82]. Altogether, these data support the use of upfront surgery with risk-based addition of adjuvant therapy in patients with HPV-negative locally advanced OPSCC [10].

HPV-positive OPSCC occurring in smokers (>10 pack-years) exhibit an intermediate prognosis between HPV-positive tumors in non-smokers, which are associated with the best prognosis, and HPV-negative tumors, which are associated with the worst prognosis [6,83,84]. Interestingly, in a recent prospective nonrandomized longitudinal study on 279 patients with OPSCC, Seikaly et al. showed that primary surgery offered the best survival outcomes, in comparison with definitive RT with or without CT, in smokers with p16-positive OPSCC and in patients with p16-negative cancers, whereas there was no survival advantage in non-smokers with p16-positive tumors [84]. Although these results have to be reinforced by larger studies, they support the use of primary surgery followed by adjuvant therapy in smokers with HPV-positive OPSCC, in particular if surgery is feasible with minimal morbidity. Figure 2 summarizes the therapeutic strategy for patients with locally advanced resectable OPSCC.

### 4.3. Locally-Advanced Unresectable OPSCC

In addition to oropharyngeal tumors invading the pterygoid process, the skull base, the nasopharynx, or the carotid artery, those with a large tongue base involvement crossing the midline have to be classified in this category because primary surgery would lead to unacceptable functional impairment (definitive enteral nutrition, unintelligible speech) [10,85].

Definitive cisplatin-based concurrent CRT is the gold standard treatment for locally advanced unresectable OPSCC, whatever the HPV-status [8,86].

Patients with HPV-negative unresectable OPSCC bear a poor prognosis with reported 5-year OS rates inferior to 35% [69,86,87]. As mentioned previously, intensifying the therapeutic strategy by adding induction CT before CRT demonstrated no survival advantage, despite a possible benefit in terms of distant metastasis in patients with large or multiple node metastases [67].

At the opposite, even with a T4 unresectable tumor, HPV-positive OPSCC patients display satisfactory survival rates, particularly if they are non-smokers [88,89]. However, there is no alternative to the conventional cisplatin-based concurrent CRT in patients who are fit enough to receive this CT [88,89]. Indeed, in a recent retrospective review of 93 consecutive patients who underwent definitive CRT for HPV-positive OPSCC with clinical T4 disease, Bhattasali et al. found 3-year OS and DSS rates of 79% and 86%, respectively, and showed that, on multivariable analysis, the only prognostic factor was the CT regimen [88]. In a randomized, multicenter, non-inferiority trial of RT plus cetuximab or cisplatin in HPV-positive OPSCC (NRG Oncology RTOG 1016), Gillison et al. found that cisplatin-based CRT was associated with higher OS rate (estimated 5-year OS was 77·9%, 95% CI: 73.4–82.5, in the cetuximab group versus 84·6%, 95% CI: 80.6–88.6, in the cisplatin group). Of note, in the subgroup of patients with T4 and/or N3 disease treated by RT plus cisplatin, estimated 5-year OS was 66.1% [89].

### 4.4. Recurrent and/or Metastatic OPSCC

Treatment of recurrent and/or metastatic OPSCC is similar to that of other recurrent and/or metastatic HNSCC. There is no specific recommendation for oropharyngeal tumors and no particularities according to the HPV status [8,44,90]. However, even with a metastatic disease, HPV-positive OPSCC patients still harbor a better prognosis than HPV-negative OPSCC patients [90,91].

Briefly, salvage surgery, when feasible, is the gold standard therapy for loco-regional recurrence [33,92]. Whereas salvage neck dissection produced relatively favorable oncologic outcomes for nodal residual or recurrent disease, salvage surgery of local recurrences is associated with poor oncologic outcomes, particularly in HPV-negative OPSCC, and with substantial functional impairment [33,92]. (C)RT can be used in patients who did not receive RT before, if salvage surgery is not feasible or as adjuvant therapy after surgical salvage. Reirradiation of previously irradiated tumor sites can be delivered in highly selected cases [93].

Local therapy (surgery, stereotactic RT) is a valid therapeutic option for patients with a single metastatic or oligometastatic disease [94]. In other cases, systemic therapies will be delivered according to previous treatments received, general health status (PS) and comorbidities, tumor spread and patient symptoms, and PD-L1 tumor expression. The combination of cisplatin and cetuximab with 5-fluorouracil (EXTREME) or docetaxel (TPEx) are two standard systemic therapy regimens [91]. Alternatively, pembrolizumab (anti-PD1) can be used (alone or with CT: cisplatin + 5-FU) as first-line therapy since the phase III randomized study demonstrated improved OS with pembrolizumab alone in patients whose tumor expresses PD-L1 (combined positive score: CPS ≥ 1) or with pembrolizumab + CT independently of PD-L1 tumor expression [95]. In this regard, immunotherapy has been reported to be potentially more effective in HPV-positive patients, but its molecular mechanism is still unclear. However, to date, the HPV status has no impact on the indications for immunotherapy in patients with recurrent/metastatic OPSCC.

## 5. Current Research and Future Directions

The three main perspectives in the treatment of OPSCC can be summarized as follows: 1—to determine the optimal therapeutic strategy between primary surgery and RT alone in patients with early-stage OPSCC; 2—to improve oncologic outcomes in patients with HPV-negative locally advanced OPSCC with an intensified therapeutic regimen that does not raise acute and chronic treatment-related toxicities; 3—to reduce long-term functional impairment and improve QoL in HPV-positive locally advanced OPSCC patients through de-escalation therapeutic strategies without compromising survival.

As mentioned previously, primary surgery and definitive RT lead to satisfactory and comparable oncologic outcomes in patients with early-stage OPSCC [15,72,73,96]. The main goal for these patients is to reduce treatment-related morbidity in order to maintain QoL. Several prospective studies are currently ongoing with the objective to compare these two therapeutic approaches in terms of swallowing function. For example, the currently ongoing randomized phase III «best of» study (ClinicalTrials.gov Identifier: NCT02984410) compares IMRT vs. TORS in patients with T1–T2, N0–N1 OPSCC with patient-reported swallowing function at 1 year as the primary end point.

Approximately half of patients with HPV-negative locally advanced OPSCC will present tumor recurrence and will die from their cancer [15]. There is still, therefore, a crucial need to improve oncologic outcomes. Since maximum tolerable toxicity level is already reached with CRT, it is unlikely that adding conventional therapy to standard treatment will result in improved patient outcomes. Several attempts have been made in this direction but have failed to improve survival or have led to unacceptable toxicity [67,68]. In this context, there are two credible options to intensify the therapeutic approach. The first one would be to combine with conventional CRT a new therapeutic agent without cross-toxicity but that is able to potentiate the anti-tumor effects of concurrent CRT. Preliminary results of the phase III JAVELIN head and neck 100 trial have shown no benefit in terms of PFS of Avelumab (anti PD-L1) plus CRT followed by Avelumab maintenance vs. CRT despite similar tolerability [71]. The currently recruiting phase III NIVOPOSTOP study evaluates the addition of nivolumab (anti PD1) to CRT as adjuvant therapy after primary surgery (clinicalTrials.gov Identifier: NCT03576417). What is probably more promising is the phase III randomized study comparing Debio 1143 (Xevinapant), an antagonist of apoptosis proteins inhibitor, combined with cisplatin-based CRT vs. CRT alone, with event-free survival as the primary end-point (clinicalTrials.gov Identifier: NCT04459715) [97]. The second one would be to replace concurrent cisplatin by a new combination of innovative therapeutic agents. This is the case of the phase III randomized GORTEC-REACH study that compared the combination of avelumab, cetuximab, and RT with standard of care CRT [98]. However, preliminary results of this as yet unpublished study demonstrated that this new combination did not improve outcomes in patients fit to receive cisplatin-based CRT [98].

Given the favorable prognosis of HPV-positive OPSCC patients, several studies have investigated the possibility to reduce treatment intensity without compromising oncologic outcomes (treatment de-escalation) [38,99,100]. As already mentioned, replacement of cisplatin by cetuximab in association with definitive RT showed decreased survival outcomes without any benefit in terms of toxicity (de-escalate study) [66]. Since most treatment failures in HPV-positive patients correspond to distant metastasis while loco regional control is achieved in most patients, it is not logical to decrease the intensity of the systemic treatment. More promising would be to reduce the RT dose intensity. This approach would be interesting in the context of adjuvant (C)RT to take benefit from an upfront surgery that would be followed by a less toxic adjuvant treatment compared with definitive CRT. In this regard, the currently recruiting phase III PATHOS study assesses whether swallowing function can be improved following TORS for HPV-positive OPSCC, by reducing the intensity of adjuvant therapeutic protocols, with 50 Gy adjuvant RT alone as the experimental arm (clinicalTrials.gov Identifier: NCT02215265). The other strategy would be to reduce the RT dose intensity in definitive CRT protocols with or without adding a systemic treatment to conventional CRT to compensate this RT dose reduction. In this regard, a recent observational study using the National Cancer Database, by Gabani et al., showed that, in HPV-positive OPSCC patients, the use of RT doses inferior to 66 Gy did not result in reduced OS compared to standard RT doses (66 to 70 Gy) [101]. The currently recruiting phase II/III NRG HN005 trial (ClinicalTrials.gov Identifier: NCT03952585) compares a standard CRT arm (full dose IMRT, 70 Gy, with cisplatin) with two de-escalation experimental arms of either reduced-dose IMRT (60 Gy) with cisplatin or reduced-dose IMRT (60 Gy) with nivolumab in non-smoking patients with T1–2, N1, M0 or T3, N0–N1, M0 (AJCC, 8th edition) p16-positive OPSCC.

## 6. Conclusions

The role of tumor HPV-status on therapeutic decision-making in OPSCC patients is not yet well defined. However, the tumor HPV status will have, in the near future, a major impact on the therapeutic management of OPSCC patients (surgical vs. non-surgical strategy, RT doses, and RT-associated therapies). There is no published randomized phase III clinical trial comparing surgical vs. non-surgical therapeutic strategies in OPSCC patients. Nevertheless, there are convergent data supporting the use of primary surgery in patients with HPV-negative OPSCC since it is associated with improved oncologic outcomes, if an acceptable functional result can be reasonably expected. In patients with HPV-negative unresectable OPSCC, cisplatin-based CRT remains the gold standard treatment since recent studies aiming at intensifying therapeutic strategy have failed to improve both oncologic and functional outcomes. In patients with early-stage HPV-positive OPSCC, surgery and RT lead to comparable survival outcomes and treatment selection should be mainly based on treatment-related morbidity and preservation of swallowing function and QoL. In patients with HPV-positive locally advanced OPSCC, upfront surgery plus adjuvant (C)RT is associated with increased morbidity and functional impairment and no substantial gain in terms of survival compared with definitive CRT. In these patients, cisplatin-based concurrent CRT remains the cornerstone of the treatment but research is being undertaken to assess new therapeutic regimens (reduction in RT doses, and other combinations of systemic therapy) in order to minimize treatment-related toxicities.

## Figures and Tables

**Figure 1 cancers-13-05456-f001:**
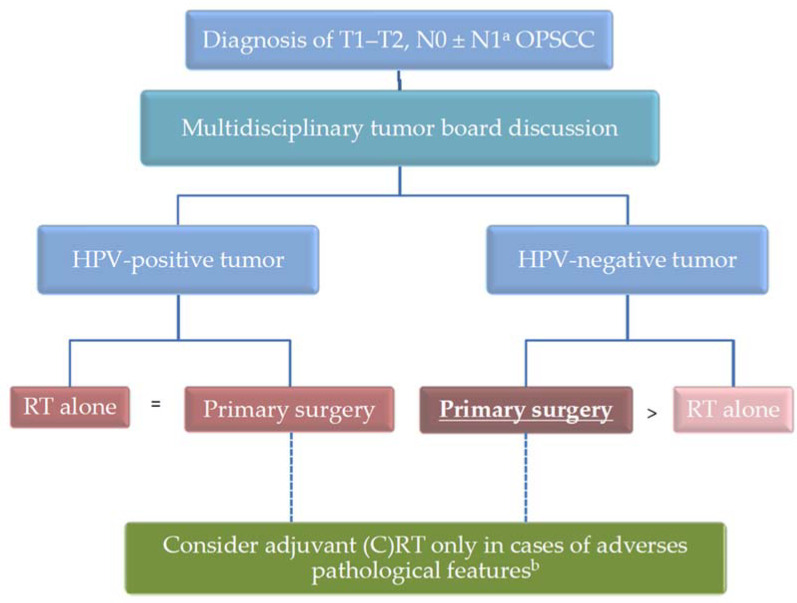
Therapeutic strategy for early-stage OPSCC. ^a^ some T1–T2, N1 OPSCC patients with a small unique ipsilateral metastatic lymph node can be classified in this category. ^b^ adverse pathological features include close/positive surgical margins, perineural or vascular invasion, metastatic lymph node(s), and extracapsular spread. The preferred therapeutic option, if any, is in bold and underlined.

**Figure 2 cancers-13-05456-f002:**
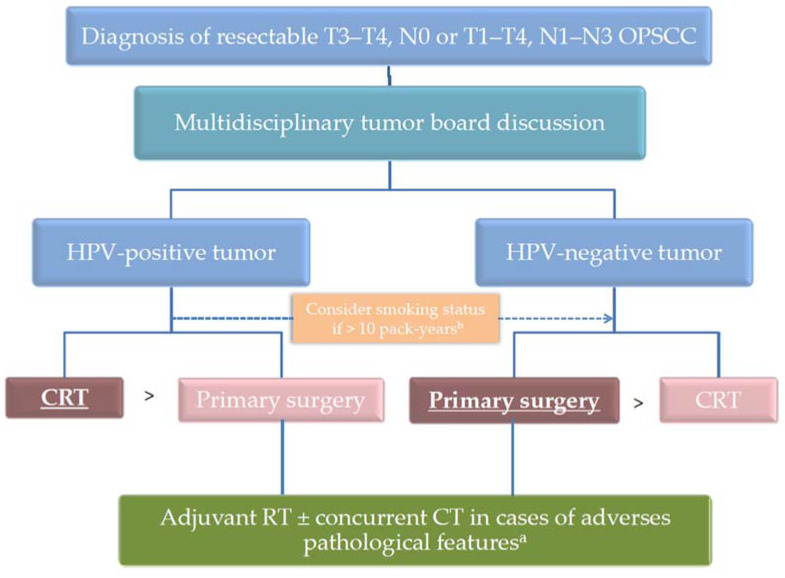
Therapeutic strategy for locally advanced resectable OPSCC. ^a^ adverse pathological features include positive surgical margins; extracapsular spread; or combination of several pejorative criteria such as close surgical margins, perineural or vascular invasion, multiple/bilateral metastatic lymph nodes. ^b^ when tobacco consumption is greater than 10 pack-years, HPV-positive OPSCC could be managed as HPV-negative OPSCC. The preferred therapeutic option is in bold and underlined. Primary surgery is only considered if an acceptable functional outcome can be reasonably expected.

**Table 1 cancers-13-05456-t001:** Main usual characteristics of HPV-positive vs. HPV-negative oropharyngeal squamous cell carcinomas (OPSCC).

Main Characteristics	HPV-Positive OPSCC	HPV-Negative OPSCC
Gender	Male >> female	Male >> female
Alcohol/tobacco	Low consumption	High consumption
General health status	Good	Poor, high comorbidity level
Educ./economic level	High	Low
Tumor location	Tongue base and tonsils	All parts of the oropharynx
Primary tumor	T1/T2, superficial/exophytic tumor	T3/T4, ulcerative and infiltrative tumor
Lymph-node involvement	Extremely frequent, multiple, cystic neck mass(es)	Moderately frequent, limited number of metastases
Second primary cancer	Very low risk	10 to 15% (head and neck, lung, esophagus +++)
Sensitivity to RT/CT	High	Variable, low to moderate
Prognosis	Good	Poor to intermediate

Educ.: educational; RT: radiation therapy; CT: chemotherapy.

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
