# Peer review of "Current Therapeutic Strategies in Patients with Oropharyngeal Squamous Cell Carcinoma: Impact of the Tumor HPV Status"

_cancers, 2021, doi:10.3390/cancers13215456_

Round 1

Reviewer 1 Report

In the review article titled “Current therapeutic strategies in patients with oropharyngeal squamous cell carcinoma: impact of the tumor HPV status”, the authors analyzed the currently available treatment options for patients with oropharyngeal SCC and discuss the impact of tumor’s HPV status on the therapeutic strategies. This is a well-written article. Following are a few comments:

- The authors frequently use CRT for chemoradiotherapy (combination of chemotherapy and radiation therapy). However, they have used the same abbreviation for chemotherapy also. For example, line 250…are authors referring to only cisplatin-based chemo or combined chemoradiotherapy? Same for line 264 and 410. Please clarify.

- Line 187-189……Citation needed.

- Line 259……the authors should briefly describe for readers what grade 3 or 4 toxicity is.

- Line 265-268…….Citation

- Please explain for readers what DSS means.

- Line 334……”Although TORS can be used in carefully selected cases”……..what do authors mean y carefully selected cases

- Line 376……..check the spelling of “negative”

- Line 448-449……….The combination of cisplatin and cetuximab……are two standard chemotherapeutic regimens. Isn’t cetuximab considered immunotherapy?

- Please rephrase lines 520-522 in conclusion section.

Author Response

Author's Reply to the Review Report (Reviewer 1)

We are grateful to the Editorial Board and to the experts for considering our submitted manuscript entitled “Current therapeutic strategies in patients with oropharyngeal squamous cell carcinoma: impact of the tumor HPV status”. We have taken into account all remarks and criticisms formulated by the reviewers and we thank the Editorial Board and the experts for their constructive comments. The changes are clearly highlighted in the text (in red) and our answers are as follows:

1- In the review article titled “Current therapeutic strategies in patients with oropharyngeal squamous cell carcinoma: impact of the tumor HPV status”, the authors analyzed the currently available treatment options for patients with oropharyngeal SCC and discuss the impact of tumor’s HPV status on the therapeutic strategies. This is a well-written article.

Author response: We thank reviewer 1 for his encouraging and positive comments.

2- The authors frequently use CRT for chemoradiotherapy (combination of chemotherapy and radiation therapy). However, they have used the same abbreviation for chemotherapy also. For example, line 250…are authors referring to only cisplatin-based chemo or combined chemoradiotherapy? Same for line 264 and 410. Please clarify.

Author response: In the manuscript, we used the abbreviation “CT” for chemotherapy and “CRT” for concurrent chemoradiation therapy. As suggested by reviewer 1, we have clarified this point by adding “concurrent” before CRT in lines 250, 264 and 410.

3- Line 187-189……Citation needed.

Author response: We have added 2 references here including 1 new reference (ref [50])

4- Line 259……the authors should briefly describe for readers what grade 3 or 4 toxicity is.

Author response: We have briefly explained what grade 3-4 toxicity is.

5- Line 265-268…….Citation

Author response: We have added 1 reference here (ref [68])

6- Please explain for readers what DSS means.

Author response: DSS (disease specific survival) was defined before (see end of the paragraph “3.1.2. Therapeutic strategies involving upfront surgery”)

7- Line 334……”Although TORS can be used in carefully selected cases”……..what do authors mean by carefully selected cases

Author response: We meant that most patients with a locally advanced OPSCC will require an open surgical approach if surgery is selected as the primary treatment modality but that, however, there is still a small role for TORS in this situation. We have clarified this sentence by adding some details on the patient selection criteria.

8- Line 376……..check the spelling of “negative”

Author response: We have corrected it

9- Line 448-449……….The combination of cisplatin and cetuximab……are two standard chemotherapeutic regimens. Isn’t cetuximab considered immunotherapy?

Author response: For more clarity, we have replaced “chemotherapeutic regimens” by “systemic therapy regimens”.

10- Please rephrase lines 520-522 in conclusion section.

Author response: We have rephrased the two first sentences.

Reviewer 2 Report

In this review, the authors cover the therapeutic strategies currently used to treat OPSCC, with particular focus HPV positive tumors. The difference in HPV+ and - tumors has been well discussed with regards to clinical presentation and response to treatment, and this is well summarised.  This is a comprehensive and well-written review, covering both surgical and non-surgical treatment options for OPSCC, at different stages of disease. 

Author Response

We are grateful to the Editorial Board and to the experts for considering our submitted manuscript entitled “Current therapeutic strategies in patients with oropharyngeal squamous cell carcinoma: impact of the tumor HPV status”. We have taken into account all remarks and criticisms formulated by the reviewers and we thank the Editorial Board and the experts for their constructive comments. The changes are clearly highlighted in the text (in red) and our answers are as follows:

1- In this review, the authors cover the therapeutic strategies currently used to treat OPSCC, with particular focus HPV positive tumors. The difference in HPV+ and - tumors has been well discussed with regards to clinical presentation and response to treatment, and this is well summarized.  This is a comprehensive and well-written review, covering both surgical and non-surgical treatment options for OPSCC, at different stages of disease.

Author response: We thank reviewer 2 for his encouraging and positive comments.

Reviewer 3 Report

This is a timely review on a topic of great interest. Authors adequately summarized the current knowledge on HPV-driven OPSCC and comprehensively presented the therapeutic options for early stage, locally advanced resectable, locally advanced unresectable and recurrent and/or metastatic OPSCC, discussing the impact of HPV tumor status.

Minor criticisms:

  • Line 81: this is a very simplistic and not accurate explanation of p16 over-expression; please see McLaughlin-Drubin et al, 2011 for an updated explanation, since induction of p16INK4A by HPV16 E7 is not dependent on pRB inactivation; re-phrase the sentence
  • Lines 97-98: “The frequent cystic morphology of neck metastasis in HPV-related OPSCC can suggest a congenital neck cyst”; please clarify
  • Line 106: I would suggest to replace “more likely to receive optimal treatment” with “more likely to benefit from the treatment”
  • Lines 126-128 (“Despite the HPV oncogenic properties at other cancer sites and in particular the anogenital organs, the risk of a second HPV- induced primary malignancy seems relatively low and does not represent an important cause of death in HPV-related OPSCC patients”): I am not sure about the meaning of this statement. HPV does not cause a systemic infection. An HPV infection at the level of H&N does not mean that the patient is also infected at the ano-genital tract. Oral and ano-genital infections are acquired independently.
  • Lines 351 and 353: I believe that % is missing for the rates
  • Line 518: it seems the end of the sentence is missing (“in patients”…?)

  • Be consistent in the use of HPV-related or HPV-induced or HPV-positive when referring to OPSCC caused by HPV
  • Be consistent in the use of T1–T2, N0 or T1–T2N0 when referring to early stage
  • Minor English errors (e.g., line 154 and 389: “worse” instead of “worst”; line 376 “p16-negaticve” instead of “p16-negative”)
  • Table 1. Re-phrase the title since clinical characteristics are shown along with others regarding sociodemographics (gender) and lifestyle (alcohol/tobacco). What do “+++” indicate in this Table?
  • Figure 2. In light of the data presented in the respective section, it would be interesting to distinguish patients with HPV-driven tumors based on the smoking status, even though the best therapeutic strategy for smokers vs. non-smokers has not been identified yet; this could still be shown in the Figure
  • Ref 30, 34 and 95 are incomplete

Author Response

Author's Reply to the Review Report (Reviewer 3)

We are grateful to the Editorial Board and to the experts for considering our submitted manuscript entitled “Current therapeutic strategies in patients with oropharyngeal squamous cell carcinoma: impact of the tumor HPV status”. We have taken into account all remarks and criticisms formulated by the reviewers and we thank the Editorial Board and the experts for their constructive comments. The changes are clearly highlighted in the text (in red) and our answers are as follows:

1- This is a timely review on a topic of great interest. Authors adequately summarized the current knowledge on HPV-driven OPSCC and comprehensively presented the therapeutic options for early stage, locally advanced resectable, locally advanced unresectable and recurrent and/or metastatic OPSCC, discussing the impact of HPV tumor status.

Author response: We thank reviewer 3 for his encouraging and positive comments.

2- Line 81: this is a very simplistic and not accurate explanation of p16 over-expression; please see McLaughlin-Drubin et al, 2011 for an updated explanation, since induction of p16INK4A by HPV16 E7 is not dependent on pRB inactivation.

Author response: We have provided an updated explanation of p16 over-expression and we have included this new reference in the bibliography (McLaughlin-Drubin et al, 2011).

3- Lines 97-98: re-phrase the sentence “The frequent cystic morphology of neck metastasis in HPV-related OPSCC can suggest a congenital neck cyst”; please clarify.

Author response: We have rephrased this sentence.

4- Line 106: I would suggest to replace “more likely to receive optimal treatment” with “more likely to benefit from the treatment”.

Author response: We have done this correction.

5- Lines 126-128 (“Despite the HPV oncogenic properties at other cancer sites and in particular the anogenital organs, the risk of a second HPV- induced primary malignancy seems relatively low and does not represent an important cause of death in HPV-related OPSCC patients”): I am not sure about the meaning of this statement. HPV does not cause a systemic infection. An HPV infection at the level of H&N does not mean that the patient is also infected at the ano-genital tract. Oral and ano-genital infections are acquired independently.

Author response: We have clarified this statement and we have added 2 new references on this topic (ref [35, 36]).

6- Lines 351 and 353: I believe that % is missing for the rates.

Author response: We have added “%” after survival rates.

7- Line 518: it seems the end of the sentence is missing (“in patients”…?).

Author response: We have completed this sentence.

8- Be consistent in the use of HPV-related or HPV-induced or HPV-positive when referring to OPSCC caused by HPV.

Author response: We have done the requested corrections. We have decided to use “HPV-positive or HPV-negative OPSCC”.

9- Be consistent in the use of T1–T2, N0 or T1–T2N0 when referring to early stage

Author response: We have done the requested corrections. We have chosen “T1–T2, N0”.

10- Minor English errors (e.g., line 154 and 389: “worse” instead of “worst”; line 376 “p16-negaticve” instead of “p16-negative”)

Author response: We have done these corrections.

11- Table 1. Re-phrase the title since clinical characteristics are shown along with others regarding sociodemographics (gender) and lifestyle (alcohol/tobacco). What do “+++” indicate in this Table?

Author response: We have suppressed the term “clinical” from the title. We have replaced “comorbidities +++” by “high comorbidity level”.

12- Figure 2. In light of the data presented in the respective section, it would be interesting to distinguish patients with HPV-driven tumors based on the smoking status, even though the best therapeutic strategy for smokers vs. non-smokers has not been identified yet; this could still be shown in the Figure

Author response: We have modified Figure 2 accordingly.

13- Ref 30, 34 and 95 are incomplete

Author response: We have completed these three references.

Reviewer 4 Report

Thank you for the opportunity to review the manuscript : “Current therapeutic strategies in patients with oropharyngeal squamous cell carcinoma: impact of the tumor HPV status” by  Bozec A, et al.

Comments to the Author
In this manuscript, the authors review the impact of HPV in the biology, prognosis and treatment of oropharyngeal squamous cell carcinoma, as well as the different therapeutic approaches to this disease. Finally, they provide treatment strategies on this regard. The review is well written and comprehensive. I do not have any relevant concern.

Regarding self-citation, reference number 51 is probably not needed.

Author Response

Author's Reply to the Review Report (Reviewer 4)

We are grateful to the Editorial Board and to the experts for considering our submitted manuscript entitled “Current therapeutic strategies in patients with oropharyngeal squamous cell carcinoma: impact of the tumor HPV status”. We have taken into account all remarks and criticisms formulated by the reviewers and we thank the Editorial Board and the experts for their constructive comments. The changes are clearly highlighted in the text (in red) and our answers are as follows:

1- In this manuscript, the authors review the impact of HPV in the biology, prognosis and treatment of oropharyngeal squamous cell carcinoma, as well as the different therapeutic approaches to this disease. Finally, they provide treatment strategies on this regard. The review is well written and comprehensive. I do not have any relevant concern.

Author response: We thank reviewer 4 for his encouraging and positive comments.

2- Regarding self-citation, reference number 51 is probably not needed.

Author response: This reference was used to illustrate the development of three-dimensional preoperative virtual planning in fibula free-flap mandibular reconstruction.